# Physiological and Behavioral Benefits for People and Horses during Guided Interactions at an Assisted Living Residence

**DOI:** 10.3390/bs11100129

**Published:** 2021-09-23

**Authors:** Ann Linda Baldwin, Barbara Kathleen Rector, Ann Calfee Alden

**Affiliations:** Department of Physiology, University of Arizona, Tucson, AZ 85721, USA; barbarakrector42@gmail.com

**Keywords:** horse–human interaction, equine-assisted learning, heart rate variability, diastolic pressure, social bonding

## Abstract

Assisted living is a fast-growing living option for seniors who require residence-based activities for maintaining mental and physical health. Guided equine interactions may benefit seniors, so an on-site equine program was started at Hacienda at the River senior living community. For research purposes, twenty-four residents and associates, aged fifty-five or over, consented to physiological measurements before, during and after four guided sessions of stroking one of three horses for 10 min over 4–6 weeks. Heart rate variability (HRV) was measured simultaneously in humans and horses during interactions. We hypothesized that human heart rate (HR) and HRV would increase during stroking and HRV power would shift toward the very low frequency (VLF) range common in horses, indicative of healthy function. During stroking, human HR increased (*p* < 0.05) but HRV (SDRR) and %VLF of HRV power did not change. Diastolic blood pressure (DBP), an exploratory measure, significantly increased after stroking, consistent with arousal. Two horses showed no significant changes in HR or HRV, but one relaxed. Sixteen horse–human pairs demonstrated synchronized HRV peak frequencies during sessions, suggestive of social connection. Participants used more positive than negative words describing their experience during exit interviews (*p* < 0.05). These data show that horses animate seniors without causing emotional stress and provide opportunities for social bonding.

## 1. Introduction 

Assisted living is one of the fastest-growing sectors of living options for seniors in the USA. According to the U.S. Census Bureau, the percentage of population over the age of 65 is expected to reach more than 20% by 2030. The choice of assisted living is popular because it provides senior citizens with companionship, security, and assistance with daily activities. Many residents in assisted living experience one or more of the following conditions: (i) limited mobility, (ii) chronic health problems, such as heart disease and diabetes, and/or (iii) diminished cognition. In order to maintain physical, mental and emotional health, it is essential that seniors have access to a variety of residence-based stimulating activities [1]. 

### 1.1. Equine-Assisted Learning and Heart Rate Variability

One activity that has recently been shown to benefit the middle-aged and elderly is equine-assisted learning (EAL). During EAL, participants interact with horses, under the guidance of trained professionals, to develop life skills and achieve educational and personal goals. A recently published study indicated that EAL can improve cardiovascular function in healthy, independently living individuals over the age of fifty-five [2]. Participants focused on their bodily sensations and observed the responses of the horse as they moved towards and around the horse. During this activity, their HR and heart rate variability (HRV) significantly increased. Heart rate variability, or changing of the HR with time, is enhanced by a rhythmic exchange between stimulation of the sympathetic and parasympathetic branches of the autonomic nervous system (ANS); this allows the heart to adapt to physical and emotional requirements, leading to reduced stress and anxiety, improved mental and physical performance and a more synchronized connection between the heart and the brain [3]. Higher HRV is correlated with improved cardiovascular function [4], emotional wellbeing [5,6,7] and social engagement [8]. The increase in participant HR and HRV observed during EAL in the study by Baldwin et al., 2018 [2] suggests that guided equine interactions animate older people without causing them emotional stress. In addition, when participants in this study interacted with the horses, their HRV tended to shift towards the very low frequency (VLF) portion of the frequency spectrum where the HRV of horses predominates. In humans, low levels of VLF are strongly correlated with death after a heart attack [9], metabolic syndromes [10] and chronic inflammation [11,12]. Therefore, shifts to higher levels of VLF could be beneficial in some cases. Less is known about factors regulating the VLF domain than for the low and high frequency domains of HRV, but previous research has shown that sympathetic neural activity decreases the VLF band, and vagal activity increases the VLF band [13]. So far, there have been no investigations of the effects of EAL on the ANS of seniors in assisted living.

### 1.2. Equine-Assisted Learning and Social Bonding

Working with horses requires non-verbal communication skills. As humans become more aware of their own body sensations and better able to attune to the horse’s body to understand what is being communicated to them, the horse and human are able to engage in a two-way conversation, which may in time result in social bonding [14]. Recent studies have indicated that social and behavioral bonding can be linked to physiological synchrony between individuals, specifically regarding heart rate variability. This is important because physiological synchrony may contribute to the formation of group cohesion and coordination. Gordon et al., 2020 [15] showed that when three-person groups performed synchronized drumming, the members’ HRVs synchronized to a much greater extent than for groups performing asynchronized drumming. It was also found that physiological synchronization predicted individuals’ experience of group cohesion. In the Baldwin et al., 2018 [2] study, seven out of twenty-four participants showed an exact match of one or more of their HRV frequency peaks with that of their horse, also suggesting a physiological coupling. In addition, participants experienced improved self-esteem following the interaction. Adults in mid to later life are particularly prone to psychological disorders such as anxiety, depression, and low self-esteem, and they sometimes may experience social withdrawal. If EAL can promote social bonding between horse and human as manifested by pleasurable sensed experiences and possible physiological coupling, older people would benefit greatly. Baldwin et al. noticed in their previous study [2] that participants who bonded with their horse were eager to share their experience with others, encouraging social interaction. Opportunities for social interaction are essential for the elderly because those who lack the brain stimulation that human interaction provides are at greater risk of cognitive decline [16].

### 1.3. Equine-Assisted Learning in Assisted Living Facilities

There is little research on the benefits of EAL as applied to older people, and only a few studies have focused on seniors in assisted living. Although the published investigations are promising regarding the reported benefits of EAL administered in assisted living facilities, they all required the participants to leave their place of residence to attend each session. In addition, very few quantitative data were recorded, and no measurements were made on the horses, so it was impossible to investigate whether synchronistic responses occurred between each horse–human pair.

Two published studies illustrate the status of research on the physiological and psychological effects of EAL on older adults in day care centers. One experiment [17] involved participants over sixty years old with functional or cognitive limitations. Half participated in weekly equine-assisted psychotherapy (grooming, feeding, exercise—leading and lunchtime) for 6 weeks and the others did their usual activities. After a short break, the two groups switched. Outcome parameters were based on questionnaires and interviews and so were only qualitative in nature. It was concluded that the participants found meaning and purpose from their interactions with the horses. The authors recommended that future research should explore which components of therapy are most beneficial to older adults with cognitive impairments. To address that question, the present study investigates *Hand Grooming* (stroking a horse with one or both hands) as a potential candidate and uses quantitative as well as qualitative outcome measures to ensure objectivity.

The second study [18] focused on an equine-assisted intervention for people with Alzheimer’s disease/dementia who attended an adult day care center. Half the participants were assigned to weekly equine-assisted psychotherapy (grooming, observing/leading, painting horses) for 4 weeks at a farm away from the center while the others did their usual activities at the center; then the two groups switched. Trained research staff recorded and assessed the behavior and mood of the participants during the interactions, based on words and gestures used during the interactions. On the days that the participants were scheduled to work with horses, behavioral problems decreased. Human salivary cortisol concentration was elevated after the interaction in the higher mental functioning participants, suggesting an increased level of arousal. The authors interpreted this response as participants experiencing a sense of exhilaration without physiological damage. They concluded that the benefits could result from the enriched environment of the farm rather than directly from the horses. For that reason, the present study was performed in the urban setting where the participants resided or worked.

A third study [19] involved single veterans and veteran couples who spent 5 days with two equine facilitators at a rural equine therapy property. The activities included grooming the horses, interacting with them while un-haltered, trail walks, meditation and group discussions. Outcome parameters were based on questionnaires given before and after the program and three months later. Participants showed fewer psychological problems and greater happiness and quality of life (QoL) after the intervention, but there was no control group. Three months later, the initial reduction in psychological symptoms was only sustained in couples

### 1.4. Limitations of Previous Studies

A major limitation of two of the three previous studies [17,19] is the reliance on self-reported information from the participants rather than objective data. In addition, one study [19] did not include a control group. Although the questionnaires were validated and interesting details were obtained from the interviews, inclusion of an objective physiological measure would have considerably strengthened the validity of the conclusions. Quantitative outcome parameters are not influenced by acquiescence bias, in which the participant tends to agree with whatever is being asked, or confirmation bias, in which the participant would like a certain idea or concept to be true. For that reason, in the present study, we have emphasized quantitative outcome parameters.

A second problem is that all three studies involved a combination of activities, some with horses and some not, and so it is impossible to isolate the effects of the horses. These studies were not designed to determine whether the beneficial effects experienced by participants were due to the horses, the environment, or the social setting. In all cases, the participants had to travel to special equine centers, which were rurally located. There was no way to distinguish between the effects of the horses versus those of the rural environment, or just getting out of the house. The present study was completed at an urban assisted living residency and is the first in which elderly residents did not have to leave their homes to interact with the horses; seeing the horses was an integral part of their day.

A third problem that affects all types of experimental outcomes is selection bias, because the participants were not randomly selected. Only those who chose to take the opportunity to visit the horses were accepted into the study. Therefore, all participants may have been biased in terms of expecting a good result. This type of bias is difficult to eliminate because there is an inherent risk involved in working with horses and, generally, only those people who have some affinity for horses or those who crave new experiences, are willing to participate.

Finally, none of the three studies included measurement of the horses’ responses. In one of the previous studies [17], it was stated that each of the 6 horses participated for thirty or sixty minutes per session and that they showed no visible signs of stress. In horses, stress is not necessarily captured by observing their behaviors alone. By default, horses may hide their fear, anxiety, confusion or physical discomfort as those displays make them easier targets for predators in the wild. We then erroneously believe a horse is doing well in conditions that may not be natural or optimal. In the present study, we measured HRV of the horses before, during and after each interaction to determine the effects of the interactions on their ANS, and to explore whether the equine HRV response synchronized with that of their human partner, as observed in previous studies [2,20]. Such synchronization is relevant to socialization because in 2017, McCraty [21] proposed that coupling of HRV rhythms between sentient beings is a manifestation of emotional bonding.

### 1.5. Aims of the Present Study

The main aim of the present study was to test the effectiveness of an on-site equine program at a senior living community in enhancing residents’ autonomic function, body awareness and social interactions with horses and other humans. Do people over fifty-five years old in assisted living receive similar benefits from EAL as those living independently?

The following hypotheses were tested:i.Human HR and HRV will increase during the equine interaction and HRV will shift toward the very low frequency (VLF) range, indicative of healthy function.ii.Human participants will become more aware of their bodily sensations after EAL and will comment positively on their experience.

The results demonstrated that the participants became animated during EAL (increased HR and DBP) and did not experience emotional stress (no significant reduction in HRV). The absence of the hypothesized increase in participant HRV and %VLF during EAL may have been due to participants being distracted by the traffic noise in the urban setting, either consciously or subliminally. The horses, which were specially selected and trained for EAL, showed no signs of physiological stress (no significant reduction in HRV). Two-thirds of the horse–human pairs demonstrated synchronized HRV peak frequencies during sessions, suggestive of social connection between horse and human. Participants used significantly more positive than negative words when describing their sensory experiences and became more expressive and socially engaged with the facilitator and interns in subsequent sessions. These data demonstrate that a specific on-site EAL program at a senior living community allows horses to animate the resident and affiliated seniors without causing emotional stress and encourages the seniors to form social bonds with the horses and other humans.

## 2. Materials and Methods

### 2.1. Overview

This study took place between May 2018 and March 2020 at an assisted living and memory care community, The Hacienda at the River, in Tucson, AZ, USA. This is a multipurpose seven-plus acre community campus that houses and cares for people in assisted living, memory care, short-term rehabilitation, skilled nursing facility and hospice. The residents have age-related issues, such as partial mobility, impaired balance, Alzheimer’s, other forms of dementia, Parkinson’s disease, stroke and multiple sclerosis, etc. A unique characteristic of this residence is that it includes a stable for horses working in the Adventures in Awareness™ experiential educational program, “In the Presence of Horses™”, a specifically designed research-based curriculum for seniors. The Professional Association of Therapeutic Horsemanship (PATH) International (www.pathintl.org, accessed on May 2021) Standards and Safety Guidelines form the foundation of the program. The assisted living campus is in an urban environment and the on-site equine accommodation is situated in the front of the main building on a busy road. Therefore, any beneficial effects observed in residents as they interact with horses cannot be ascribed to a quiet, rural environment.

### 2.2. Design

This was a single arm “repeated measures, within subjects” design in which each participant acted as their own control, measurements being made before, during and after each interaction. Participants were also tracked longitudinally over four interactions spanning 4–6 weeks. The reason for not including a control group was that the venue was the home of most of the participants and it would have been unethical to restrict some of them from visiting the horses, as would be necessary for the control group. Without a control group, it is harder to be certain that the outcome was caused by the interaction with the horse and not by other variables. However, repeated measures designs can be very powerful because they control for factors that cause variability between subjects. For this reason, fewer subjects are required to detect a desired effect size. In addition, repeated measures designs can track an effect overtime, such as the learning curve for a new task. In this study, since the participants gradually became more familiar with the heart-focused equine interaction over time, the repeated measures design was ideal because it allowed the same subject to be measured at multiple times rather than different subjects at one point in time for each.

Twenty-nine residents, family members or associates, aged fifty-five or over, consented to physiological measurements before, during and after each of the four equine interactions spaced over 4–6 weeks. Heart rate variability, blood pressure and respiration rate were measured before and after each interaction, and HRV was measured simultaneously in humans and horses during interactions. Each participant chose a facilitator to help them slow their breathing, engage with a horse and stroke the horse for 10 min with one or both hands in conjunction with mindful breathing. After each interaction, a videorecorded exit interview was conducted during which participants were asked to describe the feelings or sensations they experienced. Each session lasted about 45 min in total. The protocol was approved by University of Arizona, Human Subjects Protection Review Board and the Institutional Animal Care and Use Committee and meets the criteria for approval under 45 CFR 46.110, 45 CFR 46.111 and/or 21 CFR 50 and 21 CFR 56. The timeline for each research day is shown below.

### 2.3. Participants

#### 2.3.1. Humans

Participants were recruited by informing them of the horse program and the opportunity for participating in research by word of mouth. If they showed an interest, they or their designees were asked to complete the AIA research release form and invited to watch the protocol with the horses and have their questions answered. If they wished to participate, and fulfilled the following inclusion criteria, they, or their designee, were asked to complete the research consent form. The consented individual would then begin their participation on the next available research day.

Inclusion criteria for human participants were:Age fifty-five or over;English-speaking;Must be a resident, family member of a resident, or working affiliate at The Hacienda at the River Assisted Living;Must verbally volunteer for the study;Consent form must be signed by themselves or by their designated power of attorney;Must be acquiescent to wearing HRV monitor around chest and/or earlobe;Have no known cardiac arrhythmias;No metal plates, pacemakers or similar devices in the body to prevent possible interference with heart rate monitors.

Of the twenty-nine participants that initially consented, one dropped out due to relocation, and four were found not to meet the inclusion criteria due to irregularities in their heart rhythms. Of the twenty-four participants who completed the study, 7 were male and seventeen were female. Five participants used wheelchairs. Four participants used a walker or a cane and one was unsteady on her feet, so all these participants used the gait belt as described in the section “Monitoring for Safety”. Thirteen participants were familiar with horses from their past. Five participants took one or more prescribed medicines on a regular basis: as follows:Bumex, Spirolactalone, Telemesartan;Amlodipine;Propranolol, Furosemide;Metropolol, Furosemide;Diltiazem, Furosemide.

#### 2.3.2. Horses

Prissy: American Quarter Horse, mare, aged in 20 s, 15 hands;

Joe: Certified Wild Horse, gelding, aged thirteen years, 13.1 hands;

Herman: Arabian, gelding, aged eighteen years, 14.2 hands.

All three horses had been working in a therapeutic capacity for at least two years and met standards of health and treatment for Professional Association of Therapeutic Horsemanship (PATH) International. They had been screened for compliant, easy-going temperament, impeccable ground manners and willingness to tolerate groups of people working around them in close quarters. The horses were in their teens or twenties and were accustomed to being near people with mobility assistive devices such as walkers, wheelchairs, canes and assistants to stabilize them when standing near the horse. The horses were closely examined each day to ensure that they were fit and willing to engage in activities. All horses were actively ridden and regularly engaged in schooling and lesson programs when not at the Hacienda, providing them with healthy exercise and further interactions with people, which they seemed to enjoy. Prissy participated throughout the whole study, and Joe participated for the first ten months, after which he appeared to lose interest in interacting with the participants and was then replaced by Herman.

### 2.4. Order of Procedures

Baseline measures of participant blood pressure (BP), respiration rate (RR) and HRV in office.Introduction of equine research team.Safety agreement. All interactions with the horses begin with the whole group short form AIA Safety Agreement which may be pledged by participants together reading from the sign: “My name is XXX and I agree to be responsible to me, thus contributing to the safety of the group”.Participant and each facilitator choose a groundwork therapy horse card (www.groundworktherapy.com.au/i-amcards, accessed on March 2016) showing the head of a horse displaying a particular attitude and briefly state what they think the horse is “saying” to them. This activity acts as an icebreaker for social interactions.Participant selects the facilitator with whom they would like to work. The other facilitator acts as a safety sentinel who constantly observes the whole environment, alert for changes that may impact the safety of the humans and horses and calls a “time out” if necessary.HRV monitors for human and *both* equine participants are turned on simultaneously. The investigators and assisting physiology student interns from University of Arizona were already trained to operate the monitors and knew how to behave around horses.Facilitator guides the participant in heart breathing and directing attention to the horses, mutual choosing and *Hand Grooming*.HRV monitors for human and equine participants are turned off simultaneously and the participant and each research team member are asked for one word to describe their experience.Exit interview of participant is conducted by the chosen facilitator and videorecorded in the private screened office where the post measurements are then made.Post interaction HRV measures of both the horses and of the human participant are taken, as well as RR of the human. The post HRV measurements of the horses are also used as the pre HRV measurements for the next participant to avoid redundant measures. The time interval between equine post measures for the first interaction, and the start of the second interaction, is kept short at 5–10 min. During this time, horses are free to eat hay and drink if they wish.Post interaction BP of human is measured

### 2.5. Resources

Purpose built 20 × 30-foot equine paddock with covered porch and special equine pavers (non-slip, well drained, suitable for walkers, canes and wheelchairs), cooled main room, office and ADA compliant bathroom.

-Selection of 4 facilitators and 2 equine professionals, trained in CPR and first aid, all solid in their horse–human interaction skills, promoting both engagement and meditative peace.-Daily nurse on site.

### 2.6. Procedures

#### 2.6.1. Pre-Measures of Human Participants

Each participant was scheduled to appear at the stable office at a particular time and the baseline measurements were made on an individual basis in a closed office. All participants were instructed to wear closed toed shoes and loose, comfortable clothing. Each participant was greeted and seated for 5–10 min in the main room. The blinds were closed so that they could not see the horses outside. They were then shown into the office, seated and their blood pressure (BP) measured using a sphygmomanometer with an inflatable cuff, which was wrapped around their upper arm positioned on a table and approximately level with their heart. Next, their HRV was measured for 5 min using a Zephyr Bioharness BT, which was secured snugly around their chest, beneath the breast, in contact with their skin. An Inner Balance HRV monitor (HeartMath^TM^), placed on their earlobe, was used as a backup. Most participants were asked to stand for this measure, to match the standing position they would assume while interacting with the horse. If the participant was using a wheelchair, all measurements were made while they were seated. Participants who had difficulty standing for periods of 5–10 min were given the option of using a wheelchair. During the HRV measure, a student also recorded their respiration rate by observing the number of times their chest or abdomen rose and fell during a period of one minute.

#### 2.6.2. Pre-Measures of Horses

Prior to the equine measure process after horses arrived on site, they were turned loose in the paddock and given fifteen to twenty minutes without halters to drink, roll and shake, eat from small hole hay bags and sometimes buck and frolic. The HRVs of the two horses participating were then recorded using a Polar Equine RS800CX belt around the girth, as described previously [2] for 5 minutes as they stood, haltered, with one equine professional holding each horse’s lead rope, in the equine paddock. This technology has been shown to be accurate and statistically reliable when making HRV measurements on horses [22].

##### Interaction Protocol

Professional Association of Therapeutic Horsemanship standards and guidelines were followed in all situations including those related to mental health. The participant was escorted outside onto the covered porch where the haltered horses were waiting with the 2 equine professionals and 2 facilitators. Often, several other residents were seated on the porch in a line against the wall. Although they were not directly participating in the research, it was their right to be there if they chose. At the beginning of the research, we discovered that their tendency to speak among themselves and to question the research personnel was distracting. For that reason, we asked them to function as “silent observers”; witnesses who contributed to the field of awareness by their presence alone. This was a successful solution to the problem for all concerned.

For the first thirteen study participants, the sessions were run between 9 am and 11 am. Due to logistical reasons relating to other needs of the Hacienda at the River senior living community, the sessions of the other eleven participants were held between 4 and 6 pm. Non-participating residents were not present at sessions held at the later time. Statistical analysis was performed both by treating the data as 2 separate groups, and by pooling the data to determine whether these factors affected the results.

#### 2.6.3. Detailed Description of Activities

##### Monitoring for Safety

If a participant usually used a walker to ensure their safe mobility, the walker was exchanged for a gait belt, placed around their waist and held securely by the facilitator to steady them so they could groom the horse more comfortably. Participants who used wheelchairs remained seated in their chair throughout the interaction. Wheelchair brakes were not locked and an assigned monitor was charged with repositioning the chair as necessary.

##### Heart Breathing, Directing Attention to the Horses, and Mutual Choosing

The facilitator directed the participant to breathe more slowly and deeply than normal, while sending feelings of love and appreciation (heart beam) towards the horses:

“Please focus your attention on the area of your heart. As you breathe in, imagine the breath going into your heart. As you breathe out, imagine the breath is leaving your heart. Count slowly to five as you breathe in and slowly to five as you breathe out. One and two and three and four and five. Now direct your breath heart beam towards the horses and sense/feel which horse’s heart is calling to your heart. As you breathe in, imagine the breath is coming from the THAT horse’s heart to your heart. As you breathe out, imagine your breath is going from YOUR heart to your chosen horse’s heart.” (breath heart beam)

At this point, the participant selected their horse. The facilitator provided guidance with the mutual choosing process as needed. The participant usually selected the horse that was showing the most interest in them; the signal may be a look, a movement of the head, an ear cocked in their direction or an actual approach, walking toward the participant who is sending breath heart beams. In this way, the horse could decide whether or not they worked with a particular person that day. At this point, the HRV monitor of the horse *not* selected was turned off.

##### Hand Grooming

The participant approached the horse in greeting, arm outstretched, palm down. If horse turned away, the participant was coached to stop, continue deep breathing and observe the horse’s reaction. Once the horse looked at the participant or turned an ear towards them, the participant continued their approach to step forward and stroke the horse gently with one or both hands for ten to fifteen minutes. Patting the horse was discouraged. Participants were encouraged to be silent and focus on their breath and sensations. During this time, the equine professional ensured that the horse and participant were positioned safely relative to each other and respected the needs of the horse and human participant, providing momentary breaks from the grooming should the horse become restive. For data analysis, the HRV data recorded during the 5 min of *Hand Grooming* immediately prior to the last one minute of the interaction were used for comparison with the pre and post activity 5 min data. This portion was selected because it reflected the time after which the human participant had established contact with the horse and was engaged in the activity.

##### Exit Interview

After each EAL session, the participants were interviewed and asked to describe what sensations they felt in their body during the interactions. The object was to extend their degree of body awareness, thus providing opportunities to enhance their further experiences with the horses. In addition, the interviews created chances to engage in social interaction with members of the research team. During the exit interview, the questions asked of the participant were:What sensations were you aware of experiencing?At what point were you aware of these sensations?Is there anything else you briefly want to share?

### 2.7. Details of Experimental Measures

*Heart Rate Variability*, the degree of change in heart rate over time, is measured as: (i) the standard deviation of the interbeat interval (SDRR) that reflects the size of the variation, and (ii) root mean square of successive differences (RMSSD), related to parasympathetic stimulation of the ANS. The frequency of the HRV oscillations is divided into three ranges: very low frequency (VLF: 0.003–0.04 Hz) an intrinsic rhythm of the heart associated with good health, low frequency (LF: 0.05–0.15 Hz), and high frequency (HF: 0.15–0.4 Hz) [23]. The LF power (or %LF), or strength of the LF oscillation, reflects both sympathetic and parasympathetic activities (usually mainly sympathetic), and the HF power (or %HF) reflects parasympathetic activity. These HRV parameters have been shown to provide accurate assessment of ANS function in humans and in horses [23,24]. Based on our results with *Con Su Permiso* [2], as each human interacted with their selected horse during *Hand Grooming*, we expected an increase in their HR, SDRR and %VLF.

### 2.8. Exit Videos

Each participant was interviewed by their chosen facilitator. Answers to the three questions listed in the “Interaction Protocol” were recorded using a Sony HDR-CX240 video camera. The words used in exit interviews were quantitatively analyzed by two students to assess the comparative emotional effects of *Hand Grooming* as described in [2]. Briefly, the exit videos were transcribed, and any adjectives describing feelings and sensations experienced during *Hand Grooming* were highlighted. Neither student had been present at the data collection events and both were trained by a lead investigator to analyze word usage. The analysis process was validated by having both students rate the first three exit videos for words to assess inter-rater reliability. The results were almost identical. This calibration process has been repeated with three other pairs of students involved in three other related studies, further confirming the validity of the rating procedure.

#### Words

Any adjectives used to describe feelings or sensations were separated into groups based on whether they had a positive, neutral, or negative connotation. These connotations were decided based on several information sources [25,26,27]. The total numbers of positive, neutral and negative words were determined for each interview and the averages and standard deviations calculated.

### 2.9. Data Analysis

The number of subjects was chosen by performing a power analysis on preliminary data comparing HRV before and after a meditation exercise for another study in which thirty participants (aged 30–69) focused on sensations in the palms of their hands. Power was estimated assuming a standard value (95% confidence limits or α = 0.05). The calculated power for n = 20 was 0.87 and was above the sufficient statistical power of 0.8. Therefore, using n = 25–30 would ensure that the data obtained were statistically meaningful and account for attrition.

Analysis of data was performed using “Kubios HRV Standard” software program to obtain HRV parameters. For this program, the detrending/smoothing function was off, keeping the VLF rhythms in the data. Two additional parameters, PNS index and SNS index, devised by Kubios, were used with the equine data to determine how the horses’ autonomic state at rest differed from that of the average human at rest. The PNS index is based on the three parameters reflecting parasympathetic activity, mean HR, RMSSD and Poincare plot index SD1. A PNS index value of zero means that the three parameters are, on average, equal to the normal human population average. Positive and negative PNS indices show how many standard deviations the value is above or below the normal population average, respectively. The SNS index, based on the three parameters reflecting sympathetic activity, mean HR, Baevsky’s stress index and Poincare plot index SD2, is interpreted similarly.

One-way repeated measures ANOVA tests were run for HRV, followed by pair-wise testing for significant differences between pre versus during or pre versus post for horse-related activities. The paired *t*-test was run to establish differences between pre and post values of BP and respiration rate. SigmaStat software was used for the analysis. The Shapiro–Wilk normality test was run for each comparison. Almost all distributions were normal; in cases of non-normal distributions, the Wilcoxon Signed Rank test was substituted for the paired *t*-test.

## 3. Results

### 3.1. Human Results

#### 3.1.1. Blood Pressure and Respiration

Systolic blood pressure of human participants was not affected by their *hand grooming* of the horses (Figure 1a). Although systolic blood pressure was higher during the first week compared to weeks 3 and 4, both before and after the horse interaction, this difference was not statistically significant. On the other hand, participant diastolic pressure consistently increased after the horse interaction compared to baseline, for all 4 weeks, but was only statistically significant for the first two weeks (*p* = 0.024, *p* = 0.018, respectively) (Figure 1b). There was no sustained change in baseline values of diastolic pressure throughout the 4 weeks. Although the participants were instructed to breathe more slowly during their time *hand grooming* the horses, the data show that this decrease was only maintained after the interaction during weeks 1 and 2 (Figure 1c), and the change was only statistically significant for week 1 (*p* = 0.006). Numerical values for BP and respiration rate results are included in Appendix A.

#### 3.1.2. Heart Rate Variability

Human HR increased significantly compared to baseline during the interactions with horses (*p* < 0.001) (Table 1). Values shown are averages over all participants. Contrary to our hypothesis, mean SDRR and %VLF did not significantly change (Table 1). Each quantity increased in about half of the participants and decreased in the other half. When baseline data were compared between weeks 1 and 4, no significant differences were found for any parameter, indicating a lack of any long-term autonomic effects of the interaction over the weeks.

When the data were split into those from the morning and those from the evening groups, the mean values for each parameter were statistically indistinguishable from those of the pooled group, and the significant increase in HR during the interaction compared to baseline was maintained. These findings indicate that the HRV results obtained were robust and are not measurably influenced by the time of day or by the presence or absence of human “silent observers” in the paddock.

An example of a typical HRV recording from a human participant hand grooming a horse is shown in Figure 2; Figure 2a shows the full recording, and Figure 2b shows the 5-min portion (shaded), just before the final one minute, which was used for comparison with the 5 min baseline and post interaction measures in all cases. In the full recording, a wavelike pattern can be seen for the initial 2 or 3 min that is associated with the heart breathing exercise, which greatly enhances the contribution of the respiratory component to the overall HRV. Similar to our previous study [20], although the participants were instructed to continue the heart breathing while grooming the horse, it was obvious from the HRV recordings that they did not. The wavelike HRV rhythm associated with heart breathing disappears when the participant starts hand grooming the horse (at 3 min 20 s, in this case).

In our previous study, in which participants travelled to a ranch in a rural setting and mindfully groomed horses, those participants experienced significant increases in SDRR and %VLF during the interaction compared to baseline [2]. As shown in Table 1, such changes were not uniformly observed in the present study.

### 3.2. Horse Results

#### 3.2.1. Heart Rate Variability

The horses, Prissy and Herman, showed autonomic HRV physiology consistent with a lack of stress, as indicated by data in Table 2. Prissy demonstrated a significant autonomic relaxation response during the interactions with human participants compared to her baseline assessment, as demonstrated by increased SDNN, RMSSD and PNS, and decreased SNS. Prissy was chosen by participants the most often. Almost 57% of interactions involved Prissy as the equine partner. Herman remained neutral, showing little change in ANS parameters during the interactions compared to baseline. The only statistically significant difference he showed was a slightly increased HR on average during the interactions compared to post values. Joe showed a small decrease in his relaxation response during interactions with humans, which was expressed as a significant decrease in PNS index, and a reduction in SDNN and RMSSD, neither of which were statistically significant. In addition, Joe had lower baseline SDNN and RMSSD than Prissy and Herman, indicating an innately less relaxed demeanor as assessed by lower parasympathetic stimulation, probably due to his breeding (certified wild horse), or history. Joe’s somewhat diminished autonomic relaxation response to human interaction during this study may account for his apparent loss of interest in the human participants later in the study, as assessed by his being less willing than previously to approach certain participants. However, all the horses’ HRV values stayed within normal range and there was no indication that they experienced any physiological stress. In particular, all three horses maintained high levels of %VLF (close to 70% or greater) throughout the experiment, similar to values observed in those horses engaged in EAL in our previous studies [2,20,28,29].

Data from the HRV frequency domains obtained from the Fast Fourier Transform performed by the Kubios program revealed that before and/or during interactions with humans all three horses showed a combination of some of the following VLF HRV oscillation peaks (Hz): 0.001, 0.003, 0.004, 0.005, 0.008, 0.009, 0.010, 0.011, 0.012, 0.013, and 0.014, most of which were observed in other horses in our previous study [2].

#### 3.2.2. Coupling of Heart Rate Variability Frequencies between Horse and Human

An example of a pair of HRV frequency domains obtained from a human participant and a horse during the hand massage is shown in Figure 3. A frequency synchronization is seen in the VLF range.

Data from the frequency domain of the humans’ HRV data revealed that eight out of thirteen morning participants and eight out of eleven evening participants showed HRV oscillation frequencies that matched, to three or four decimal points, with those of their horses, in two or more sessions (Table 3). Sometimes, the same participant showed HRV frequency coupling with two different horses. For example, HD chose to complete two sessions with Prissy and two with Joe and showed matching frequencies with both of those horses. Three of the morning and 3 of the evening participants each showed frequency coupling in just one of their sessions. Almost all the coupling frequencies were in the VLF range (0.003–0.04 Hz). The results of horse/human frequency coupling for the participants who experienced it multiple times are shown in Table 3 for morning (a) and evening (b) sessions, respectively. The presence of a frequency coupling was independent of whether the average %VLF had increased or decreased during the interaction compared to baseline.

### 3.3. Results from Exit Interviews

#### Exit Interview Words Comparison

Just after each participant had completed their interaction with the horse, they were invited to share a “feeling” word about their experience. The most frequent words used were calm, energized, loving, flow, rhythm, harmony and spiritual.

During exit interviews, positive words that were used more than once by more than two people (number of people in parentheses) were:

Connection/Bond (11); Relaxed (10); Calm (10); Comfortable (7); Happy (6); Peace (4); Patience (4); Soothing (3); Love (3).

Widely used neutral words were:

Energy (7); Warm (4); Focused (3); Surprising (3).

Widely used negative words were:

Nervous (4); Distracted (3).

Positive words were used significantly more than negative words (Table 4). For weeks 1, 2, 3 and 4, the *p*-values were 0.00012, 0.000013, 0.0058 and 0.0039, respectively, and the corresponding effect sizes (d) were large (0.97, 1.16, 0.64 and 0.67, respectively). Although there was a decrease in positive word choice on average over the weeks, this change was not statistically significant (*p* = 0.09612). Participants appeared to show an improvement in mood after their interactions with the horses and reported increased feelings of connection to the horse. The interns who analyzed the exit interviews reported that many participants became more attentive and vocal from week 1 to 4 and their responses changed from short, clipped statements to more wordy, descriptive answers. The participants also became more expressive and showed more social engagement with the facilitator and interns as their time in the study increased.

## 4. Discussion

### 4.1. Main Findings and Importance of Results

#### 4.1.1. Hand Grooming Is an Arousing, Stimulating Activity for Seniors

This study is the first to investigate the effects of in-house EAL sessions on residents and affiliates of an assisted living community. It is also one of only a few investigations to physiologically monitor horse–human pairs simultaneously. The HR data show that during *Hand Grooming*, the human participants showed a transient significant increase in arousal. It is unlikely that the increase in HR was due to movement of the participants around the horse as they performed the hand grooming, because the movement was very slow. In addition, in another study [20], one group of participants groomed a real horse, and a control group groomed a large, plush, simulation horse. In both cases, the participants’ movements were identical. However, when participants groomed the real horse, their HR significantly increased, whereas when they groomed the simulation, it did not change. The increase in HR is unlikely to be due to fear of the horses because, on average, participants’ HRV did not change; a fear response is almost always accompanied by a significant reduction in HRV. In addition, the valence of the feeling/sensation words participants used for the “one-word description” after the EAL experience and during the exit interviews was significantly more positive than negative. This combination of physiological and psychological responses suggests that the observed ANS arousal was due to excitement rather than fear. The excitement was also reflected in their respiration rates. Although during mutual choosing of the horse, the participants were instructed to breathe slower and deeper than normal, and to continue the slow breathing during the *Hand Grooming*, the respiration rates only significantly decreased from baseline after the first EAL session and not during subsequent weeks. In fact, during later weeks, the HRV recordings (i.e., Figure 2) indicated that participants often reverted to their usual more rapid, shallow breathing when they started stroking the horse as distinct from their initial slow breathing during mutual choosing. A similar reaction was reported in a previous study, in which participant respiration rate increased during the interaction compared to baseline [2]. This response is consistent with the sensual stimulation participants experienced as they interacted closely with the horse, causing them to forget about focusing on regulating their breathing. It was noted by interns that the participants were more attentive to the breathing instructions after the first interaction, and that they paid less attention during subsequent sessions. In fact, a common response to the request to “heart-breathe” during later sessions was “Oh I know about that”.

Another physiological response consistent with increased arousal was the temporary small but significant increase in DBP observed after EAL. A similar response has been observed when healthy men inhale the fragrance of grapefruit essential oil for 10 min [30]. In that study, increases in DBP were shown to be positively correlated with increases in vascular smooth muscle sympathetic nerve burst frequency (MSNA), and in addition, average plasma concentrations of the stress hormone, cortisol, decreased. The authors concluded that the increase in DBP and the changes in MSNA induced by fragrance inhalation of grapefruit essential oil reflected increased sympathetic arousal that was not induced through a stress response. A stress response enhances both the sympathetic–adrenal–medullary axis and the hypothalamic–pituitary–adrenal axis, the latter of which causes the release of cortisol. It is possible that a similar mechanism may be involved with the participants’ response to EAL, as measured in the present study, since both EAL and inhaling grapefruit essential oil caused an increase in DBP without a stress response. Further research may reveal whether odors from horses produce similar effects on HR and DBP as observed during and after EAL.

Although the observed increase in HR was consistent with our hypothesis, the lack of a significant increase in HRV and %VLF was not. One possibility is that 5 of the seniors were taking prescribed drugs daily that could affect their cardiovascular systems and thus, act as a confounding variable. However, on examining the weekly data from those participants, no consistent pattern of reduced HRV and/or %VLF during the interaction compared to baseline was revealed. Another possibility is that that since the Hacienda equine paddock bordered onto a noisy, main street with high traffic-flow, perhaps some participants were unable to be sufficiently receptive to the horse to form a heart-to-heart connection. In order to test this hypothesis, a small post hoc study was organized with 6 of the same participants (a convenience sample, based on availability) who travelled to a ranch in a rural setting, about 30 miles south of Hacienda at the River where they repeated the same *Hand Grooming* activity but with 2 different horses and equine professionals. It was not possible to recruit a larger number of participants due to restrictions caused by the COVID-19 pandemic. Unfortunately, one participant withdrew at the last minute. In addition, there was an equipment malfunction with one other participant. As a result, complete sets of data were obtained from just 4 participants (Table 5). This low number precluded the use of a statistical analysis but some trends in the data were evident.

Similar to the results at Hacienda, all four participants at the rural ranch showed an increase in HR during *Hand Grooming*. In addition, all four participants experienced a marked increase in SDNN, and three out of four, a marked increase in %VLF, during the interaction compared to baseline values. In contrast, at their first session at Hacienda at the River, only two of these four participants demonstrated an increase in SDNN and only one, an increase in %VLF in response to the horses. One participant remarked that the experience with the horse felt the same at the rural ranch as at Hacienda and that they did not notice the environment when interacting with the horses. This person actually experienced the same HRV responses at both locations, an increase in SDNN and a decrease in %VLF. Another participant, who showed an increase in SDNN and %VLF at the rural ranch but not at Hacienda, commented that they enjoyed the quiet environment at the ranch. Although these results do not prove that the loud traffic noise at Hacienda at the River was the cause of the variable HRV response observed at that location, compared to our previous results [2,20], the fact that *all* 4 participants in our subset responded to the horses with an increase in HRV at the rural ranch, but only 2 at Hacienda, is consistent with that hypothesis. Traffic noise is known to influence HRV. One study [31] demonstrated that when people are subjected to noise and air pollution from traffic, their HRV is significantly reduced. Another study, conducted in Germany on people of an average age of 61 years, also showed that exposure to traffic noise reduced HRV, activating the sympathetic system and decreasing the parasympathetic system [32]. Although %VLF was not measured in these studies, it is known that in humans, sympathetic activity decreases %VLF and vagal activity increases %VLF [13,33], so the potentially stressful traffic noise may also have been responsible for the lack of increase in %VLF during EAL. These results suggest that in order for seniors to gain the full benefit from EAL, the horse interaction area should be located away from traffic and other urban distractions.

Even in the absence of hypothesized increases in HRV and %VLF, these findings support our hypothesis that horses can animate humans without causing emotional stress. As mentioned previously, the Global Council on Brain Health [1] reports that it is essential that seniors have access to residence-based stimulating activities so they can maintain physical, mental and emotional health. This study demonstrates that *Hand Grooming* fulfils that need.

#### 4.1.2. Horses Are Not Stressed during Hand Grooming

Similar to our previous studies [2,20], the HRV measures of the horses did not decrease during EAL, consistent with an absence of stress. Additionally, in common with more recent previous studies that have included the VLF range in their HRV analysis [2,20,28,29,34], a large percentage of equine HRV power (about 70%) was in the VLF range. The significance of the magnitude of %VLF power in horse HRV is unclear. One study showed that when horses inhaled lavender essential oil, their HRV increased, reflecting relaxation, but at the same time, their %VLF decreased [2]. Another experiment [34] demonstrated that %VLF increased when horses performed in competitions requiring jumping higher fences compared to those with lower fences. Both these investigations imply that a decrease in %VLF from the usual 70–80% observed in healthy, well-kept horses at rest is associated with increased relaxation. Regarding the present study, all three horses maintained a normal %VLF throughout the EAL activity, neither significantly increasing nor decreasing. Herman and Joe did not show significant changes in any HRV parameters, implying a state of neutrality to EAL, whereas Prissy became more relaxed during EAL (increased SDNN and RMSSD) even though %VLF did not change.

### 4.2. Equine-Assisted Learning Enhances Social Engagement for Seniors

#### 4.2.1. Relevance of Exit Interviews

During the exit interviews after the interaction with the horse, participants reported increased feelings of connection to the horse. The “feeling” words commonly used by participants in their exit interviews (Connection/Bond, Relaxed, Calm), and the qualitative behavioral reports on participants from the interns, indicate that EAL did provide an opportunity for social engagement with the horses, which became stronger with repeated sessions. Equine-assisted learning also encouraged social engagement with other participants and with the researchers. Many people in their 80 s, 90 s and 100 s are not familiar with identifying their personal feelings or speaking about them. The fact that participants became more willing to speak with the interns about their sensual experiences with the horses during subsequent weeks indicates that they were becoming more aware of their bodily sensations and less inhibited, confirming our hypothesis. Although some participants were a little nervous and/or distracted during their interactions with the horses, the exit interviews overall indicated that the experience was very positive for almost all participants.

#### 4.2.2. Coupling of Heart Rate Variability Frequencies in Horse–Human Pairs

Another indication that most of the participants were forming strong bonds with their horses(s) was that two-thirds of the horse–human pairs developed synchronized HRV peak frequencies during sessions. This is a much higher fraction than reported in previous studies [2,20], which may be because other studies only involved one interaction per horse/human pair instead of the 4 prescribed for this investigation. The synchronized HRV observation supports the hypothesis of Lanata et al. [35] that “human and horse can be considered as 2 complex systems, and when they are in visual, olfactory or physical contact they get interacting through a coupling process”. In the exploratory research by Lanata et al. [35], HRV frequency coupling between the horse and each of the eleven humans increased when horse and human were in the same stall, and hence, in visual and olfactory contact, compared to when they were in neighboring stalls. Coupling decreased, however, when they were in physical contact. The authors speculated that the decrease may have occurred because the horse and humans could have been suspicious of each other during physical contact (grooming). It is unlikely that the participants and horses were suspicious of physical contact in the present study because each interaction was guided by an experienced equine professional who focused on the comfort and safety of all involved. Scopa et al. [36] demonstrated, using HRV, that horses appear to feel more relaxed while physically interacting (e.g., grooming on the right side) with familiar handlers compared to the same task performed by someone unfamiliar, as shown by sympathovagal balance shifting to a higher vagal contribution. These results provide evidence of a horses’ capacity to individually recognize a familiar person. In our study, the horse was always accompanied by a familiar person, the equine professional, and with repeated sessions, the horses and participants also became more familiar with each other.

Another group has hypothesized that when a horse and a human experience a cardiovascular coupling during an interaction, there is also mutual coordination of emotional states [35,36]. McCraty [37] reported a similar HRV synchronization between humans that was more likely to occur if the pair had a close working relationship or lived together in a bonded relationship. More research is required to test whether horses and humans may connect and communicate through HRV synchronization.

In EAL, the presence of an experienced equine professional to guide the horse–human interaction is especially important because the existence of an attachment bond in a horse–human relationships depends on the behavior of the participant, whereas, in dogs, attachment may occur irrespective of human behavior [38]. In this study, the equine professionals focused one hundred percent on gently guiding the participants to ensure that their behaviors did not cause any safety issues and to encourage a positive experience. According to Payne et al. [39], attachment between horses and humans has received little attention in the literature. However, in the last few years, more interest in this topic has surfaced [40]. Understanding the mechanisms behind animal–human emotional bonds may lead to the development of strategies to promote mutually beneficial relationships.

## 5. Limitations of the Study

As mentioned previously, the main limitation of this study was the lack of a control group due to the fact that residents at the assisted living community could not ethically be prevented from watching and interacting with the horses if they chose. The purpose of a control group is to eliminate confounding variables and thus, maximize the probability that any changes in outcome parameters are produced only by the experimental condition, in this case stroking the horse. Some possible confounding variables were the presence or absence of other residents watching the interactions, daily prescribed medications taken by a minority of participants, familiarity or not with horses, and sitting or standing during the measurements due to mobility issues. All these variables are consequences of the characteristics of the elderly population being studied.

It could be argued that the participants were not randomly selected because 2 of them had formed a relationship with their favorite horse prior to the study and were eager to participate in the experiment. However, the remainder of the participants had either only occasionally visited the horses or had not shown any interest at all before being approached to consent for the study. In fact, for about half of the participants, it took several rounds of recruitment efforts to gain their interest. Therefore, most of the participants can be considered not to be self-selected.

A second limitation was that only 3 horses were used in the study. However, the results obtained from these 3 horses were similar to those obtained from horses in other studies addressing EAL [2,20,28].

A third limitation was that the use of gestures during the exit interviews was not analyzed as in the past [2,20]. Non-verbal communication often provides more information than words alone. The reason that gestures were not analyzed in this case was that for many of the participants, this was their home and they were often holding items in their hands during the interviews, such as water bottles or handkerchiefs, which would interfere with natural gestures.

A fourth limitation was the need to hold sessions in the morning for one half of the participants and in the late afternoon for the other half, when other residents were absent. This limitation turned out to be to our advantage because analysis showed that the data were robust enough to be unaffected by time of day or the presence or absence of human onlookers. In summary, although the protocol could not be tightly controlled as in a laboratory situation, and some more subtle findings could have been missed, meaningful statistically significant results were obtained in a real-life situation.

## 6. Future Research

During this study, it was noted that EAL motivated some participants to engage in voluntary physical activity, such as rising out of their chair without assistance and walking up steps or ramps. Future studies could be focused on people with limited mobility to investigate if the spontaneous behavior changes shown during EAL result in greater health improvements than routine exercises without horses.

Since this study showed that connection and social bonding were important components of EAL in an assisted living community, there is a need to determine whether enough bonding is achieved through occasional encounters. The next step would be to determine whether repeated occurrences of HRV frequency coupling foster improvements in the physical, mental and emotional issues most frequently experienced by people attending EAL sessions.

## Figures and Tables

**Figure 1 behavsci-11-00129-f001:**
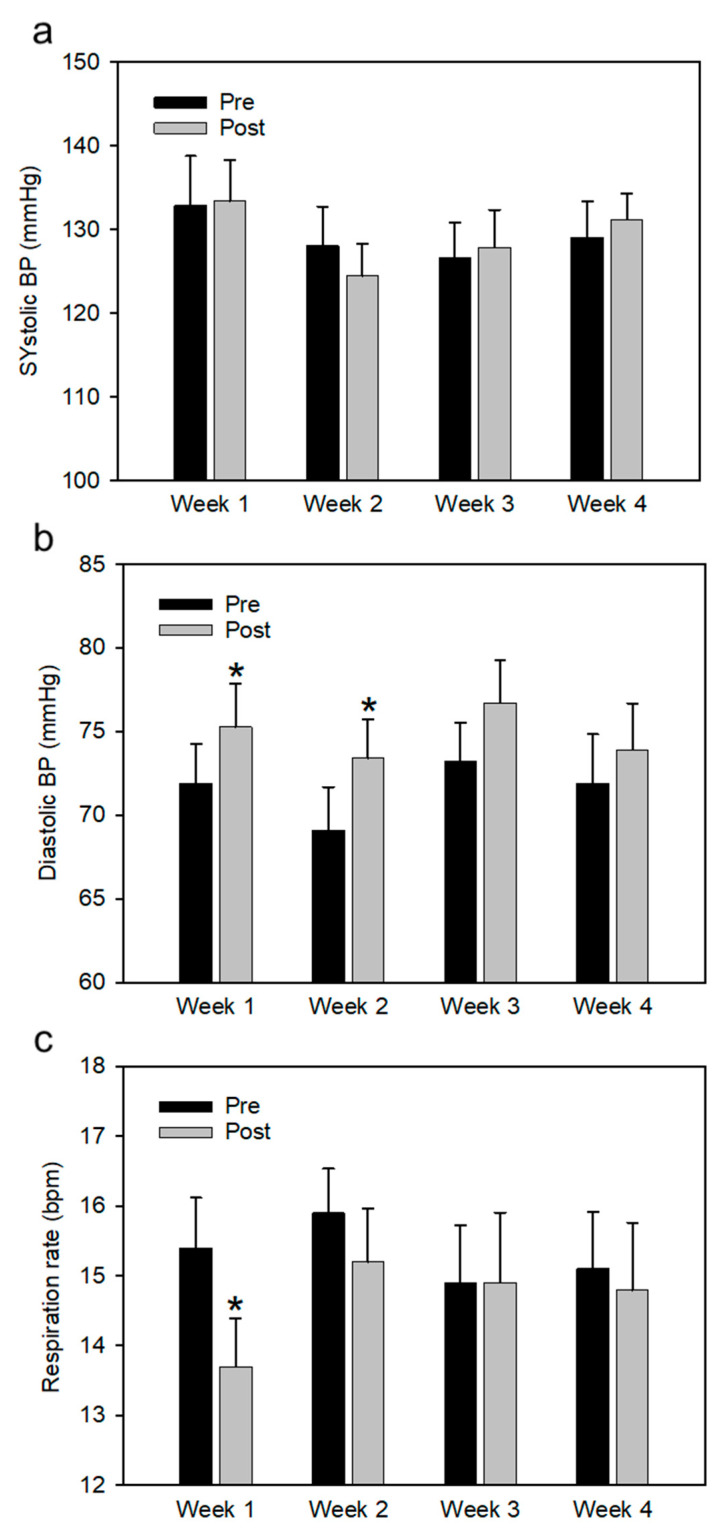
Effects of *Hand Grooming* horse (pre and post measures) on average blood pressure (**a**), diastolic blood pressure (**b**) and respiration rate (**c**) averaged over 24 participants for each week. * Denotes that result is significantly different from baseline value. Error bars denote SD.

**Figure 2 behavsci-11-00129-f002:**
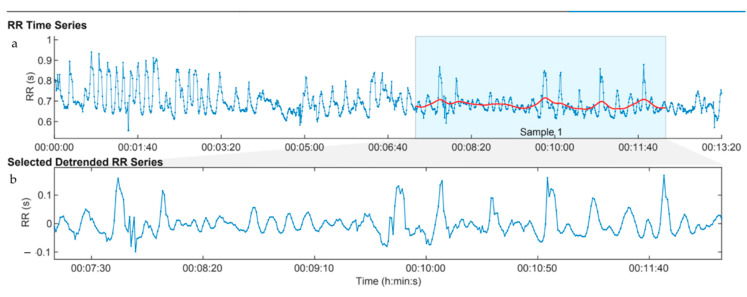
Example of heart rate variability of human participant during *Hand Grooming.* The RR time series (RR interval) represents the time interval between heart beats, which is the reciprocal of heart rate. An increase in RR is equivalent to a decrease in heart rate. (**a**) Full recording, (**b**) Just the shaded portion from (**a**).

**Figure 3 behavsci-11-00129-f003:**
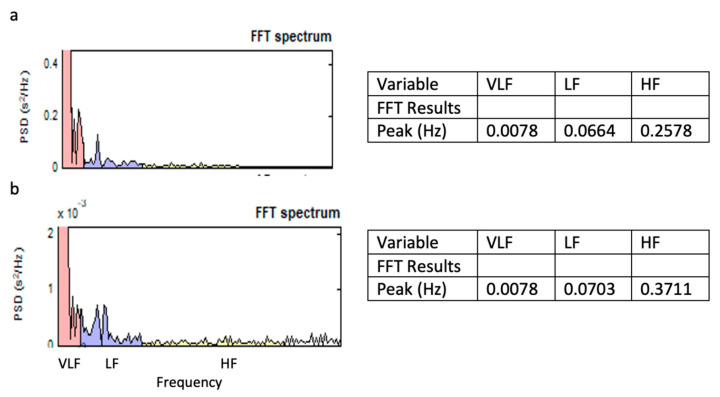
Example of human–horse heart rate variability frequency synchronization. (**a**): Fast Fourier transform frequency spectrum for human. (**b**): Fast Fourier transform frequency spectrum for horse. Note: PSD is power spectral density. Identical peaks in VLF range for human and horse.

**Table 1 behavsci-11-00129-t001:** Mean heart rate and heart rate variability averaged over 24 participants before, during and after participating in equine-assisted learning exercise, *Hand Grooming* each week for 4 weeks.

WEEK 1 (n = 24)	Baseline:	During:	Post:
HR (bpm)	80.3 ± 9.1 (SD)	85.9 * ± 10.5 (SD)	81.3 ± 10.5 (SD)
SDNN (ms)	35.9 ± 15.5	36.9 ± 16.6	30.1 ± 12.1
RMSSD (ms)	26.0 ± 23.0	29.3 ± 25.0	22.8 ± 21.5
%VLF	53.3 ± 23.3	61.9 ± 19.9	52.5 ± 24.8
WEEK 2 (n = 24)	Baseline:	During:	Post:
HR (bpm)	81.3 ± 12.6 (SD)	85.4 * ± 10.3 (SD)	81.6 ± 10.8 (SD)
SDNN ((ms)	29.5 ± 15.5	29.3 ± 11.7	29.7 ± 11.5
RMSSD (ms)	21.3 ± 18.1	21.1 ± 18.5	21.2 ± 19.0
%VLF	54.1 ± 23.9	56.1 ± 21.1	56.2 ± 26.9
WEEK 3 (n = 24)	Baseline:	During:	Post:
HR (bpm)	79.9 ± 9.4 (SD)	85.7 * ± 9.6 (SD)	80.3 ± 8.3 (SD)
SDNN (ms)	31.1 ± 13.5	31.9 ± 15.3	37.3 ± 19.7
RMSSD (ms)	23.1 ± 18.1	26.3 ± 18.3	30.8 ± 24.8
%VLF	53.4 ± 24.1	49.9 ± 25.9	51.3 ± 23.9
WEEK 4 (n = 24)	Baseline:	During:	Post:
HR (bpm)	79.5 ± 10.2 (SD)	86.3 * ± 11.2 (SD)	80.8 ± 12.2 (SD)
SDNN (ms)	32.3 ± 17.0	34.2 ± 17.2	32.9 ± 17.9
RMSSD (ms)	27.4 ± 22.2	26.2 ± 20.7	21.1 ± 19.6
%VLF	53.3 ± 21.2	57.8 ± 26.0	52.4 ± 20.9

* Value significantly different from baseline (*p* < 0.001).

**Table 2 behavsci-11-00129-t002:** Mean heart rate and heart rate variability of the three horses before, during and after participating in EAL exercise, *Hand Grooming.* Results are averaged over number of times each horse chose to interact with participants.

Prissy (n = 54)	Baseline:	During:	Post:
HR (bpm)	37.45 ± 5.47 (SD)	37.53 ± 4.67	37.33 ± 3.96
SDNN (ms)	187.45 ± 57.55	219.57 *± 96.15	191.15 ± 50.61
RMSSD (ms)	97.25 ± 36.37	109.88 * ± 46.7	110.24 ± 51.23
%VLF	72.26 ± 20.42	75.14 ± 18.19	69.36 ± 20.94
PNS	4.27 ± 1.31	4.62 * ± 1.10	4.73 * ± 1.23
SNS	−2.10 ± 0.48	−2.23 * ± 0.34	−2.24 * ± 0.29
Herman (n = 28)	Baseline:	During:	Post:
HR (bpm)	33.18 ± 2.50	35.11 ** ± 5.66	32.67 ± 1.66
SDNN (ms)	158.77 ± 84.28	162.57 ± 58.8	156.74 ± 58.63
RMSSD (ms)	94.81 ± 32.22	92.36 ± 27.40	100.15 ± 30.12
%VLF	70.38 ± 14.11	70.29 ± 16.58	66.56 ± 16.18
PNS	5.12 ± 1.16	4.84 ± 1.40	5.49 ± 0.93
SNS	−2.61 ± 0.39	−2.54 ± 0.54	−2.76 ± 0.33
Joe (n = 16)	Baseline:	During:	Post:
HR (bpm)	38.13 ± 5.20 (SD)	38.13 ± 3.22	37.67 ± 3.20
SDNN (ms)	95.86 ± 51.94	75.19 ± 43.22	87.61 ± 64.40
RMSSD (ms)	64.10 ± 41.46	44.10 ± 31.18	53.30 ± 44.67
%VLF	69.30 ± 22.46	77.23 ± 17.23	77.52 ± 13.62
PNS	3.48 ± 1.64	2.87 * ± 1.31	3.17 ± 1.68
SNS	−1.98 ± 0.82	−1.51 ± 0.88	−1.57 ± 1.01

* Value significantly different from baseline (*p* < 0.05). ** Value significantly different from post (*p* < 0.05).

**Table 3 behavsci-11-00129-t003:** Human/horse pairs showing matching heart rate variability frequencies during *Hand Grooming* interactions.

**a. Morning participants**		
**Human/Horse Names**	**Matching** **Frequencies (Hz)**	**Date of Interaction**
H.D./Prissy	0.001	14 November 2018
	0.0013	28 November 2018
H.D./Joe	0.0012	23 December 2018
	0.003	05 December 2018
M.A-P./Prissy	0.014	30 May 2018
	0.013	6 June 2018
E.B./Prissy	0.05	17 October 2018
	0.08, 0.012	31 October 2018
	0.017	12 December 2018
L.M-P./Prissy	0.013	13 June 2018
	0.031	11 July 2018
I.B./Prissy	0.021	17 January 2019
I.B. /Herman	0.012	7 February 2019
M.K./Prissy	0.023	28 February 2019
	0.0094, 0.014	14 March 2019
	0.021	21 March 2019
B.T./Prissy	0.016	21 March 2019
	0.017	28 March 2019
B.T./Herman	0.005	4 April 2019
P.N./Joe	0.0012	6 June 2018
	0.012	9 May 2018
	0.026	16 May 2018
	0.012, 0.017	23 May 2018
**b: Evening participants**		
**Human/Horse Names**	**Matching** **Frequencies (Hz)**	**Date of Interaction**
C.R./Prissy	0.079	15 August 2019
C.R. /Herman	0.0028	25 July 2019
	0.037	8 August 2019
Kr.A./Prissy	0.053	16 May 2019
	0.045	13 June 2019
K.A./Prissy	0.011	14 November 2019
	0.005	24 October 2019
P.M./Prissy	0.0056	22 August 2019
	0.08	29 August 2019
	0.0026	5 September 2019
R.K./Prissy	0.011	31 October 2019
R.K. /Herman	0.023	7 November 2019
	0.0133	21 November 2019
J.C./Herman	0.006; 0.015	27 February 2020
J.C./Prissy	0.003; 0.013; 0.06	20 February 2020
	0.133	27 February 2020
I.C./Prissy	0.005; 0.016	26 September 2019
I.C./Herman	0.022	17 October 2019
N.McN./Prissy	0.008	29 August 2019
N.McN. /Herman	0.028; 0.064	5 September 2019

**Table 4 behavsci-11-00129-t004:** Analysis of word choice during exit interviews after horse interactions. Averages and standard deviations were taken from all 24 participants.

	Week 1	Week 2	Week 3	Week 4
Positive	2.174 ± 1.466 (SD) *	1.957 ± 1.065 *	1.826 ± 1.586 *	1.522 ± 1.442 *
Neutral	0.697 ± 1.105 (SD)	0.652 ± 0.775	0.478 ± 0.846	1.000 ± 1.128
Negative	0.391 ± 0.891 (SD)	0.261 ± 0.689	0.391 ± 1.076	0.478 ± 0.730

* Number of positive words significantly greater than number of negative words.

**Table 5 behavsci-11-00129-t005:** Mean heart rate and heart rate variability of 4 humans participating in equine-assisted learning exercise, *Hand Grooming* in rural setting.

	Baseline:	During:	Post:
HR (bpm)	82 ± 10 (SD)	93 ± 17 (SD)	91 ± 13 (SD)
SDNN (ms)	28.5 ± 9.0	37.1 ± 11.2	34.6 ± 9.2
RMSSD (ms)	22.2 ± 5.0	36.3 ± 13.6	35.9 ± 22.8
%VLF	25.5 ± 10.1	43.5 ± 24.3	45.8 ± 36.7

## Data Availability

Data supporting reported results can be obtained from the corresponding author at abaldwin@u.arizona.edu.

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
