# Peer review of "Physiological and Behavioral Benefits for People and Horses during Guided Interactions at an Assisted Living Residence"

_behavsci, 2021, doi:10.3390/bs11100129_

Round 1

Reviewer 1 Report

Thank you for the opportunity to review this paper. This paper presents ground breaking contribution and analysis of the horse-human interaction and theoretical exploration around emotional bonds between horse and human.

A few minor suggestions:

Line 77 – Author states ‘Elderly people who lack the brain stimulation that human interaction provides are at greater risk of cognitive decline and dementia (https://www.luther- 78 manor.org/importance-of-social-interaction/).’ – need an evidence based reference rather one from a biased source.

line 87 – font seems to decrease in size

line 92 – Author states ‘Therefore, if EAL can promote social bonding between horse and 92 human as manifested by pleasurable sensed experiences and possible physiological coupling, older 93 people would benefit greatly.’ – Perhaps authors can describe why this is helpful – e.g. promotes bonding with other humans?

There is no hypothesis stated for the synchronizing.

Line 204- clarify what ‘this research refers to  - confused as to whether the authors mean the current study or any of the many studies referred to in previous paras.

Might be useful to provide some of the critiques of different HRV measures to explain why you used the measures you used.

Table 1- Needs more information - perhaps provide more description of the measures – is it an average across all participants?

How did you account for the effect of movement – eg brushing, moving about on the HR/HRV results?

Table 4 – needs more explanation.

Also need to describe how authors determined the significance of coupling data.

Reviewer 2 Report

Physiological and behavioral benefits

It is an interesting and important topic however I do feel that the authors could trim the manuscript considerably to improve readability and highlight their results.

Detailed comments as follows:

Line 12: How many horses?
Line 12: be consistent with your figures – I suggest writing out numbers up to ten and using numerals for numbers 11 and over

Line 16: VLF – indicative of what?

Line 19: Two horses? How many horses were used in total?

Line 24: is this really equine-assisted learning? I suggest ‘equine-assisted activities’ would be more appropriate (throughout)

Lines 72-76: This belongs in the methods section

Line 77: Why is this important?

Lines 80-81: It would be easier to put all of your research questions together at the end of the introduction so your reader doesn’t need to skip around

Lines 88-91: What did Baldwin do – real EAL or patting the horse (activity)?

Lines 104-146: This feels as though it belongs in the discussion. The introduction is so long that your reader is going to lose sight of what you are doing and why. It feels like a literature review on its own. Highlight the gaps in the knowledge and direct your reader to how you propose to fill those with this research.

Line 123: This belongs in the methods section

Line 123: You need to define Hand Grooming

Line 132: Horse or human cortisol, or both?

Line 137: ‘good stress….’ Where’s the defining line between good and bad stress?

Line 151: ‘energetic connection’ – needs defining. Body language signals?

Lines 152-154: this needs to go in your limitations section, and line 156

Lines 162-164: This is interesting but I’m not sure how it relates to the current study, please explain.

Line 168: Who labelled them as ‘interesting’?

Lines 172-4: belongs in the methods section

Line 188: you need to discuss how you overcame this limitation in your discussion

Line 186: Or just getting out of the house, perhaps.

Lines 196-99: did you have ‘control’ horses that did not interact – this would have provided the information you require here

Line 206: Needs to go to the results section

Lines 208-216: This reads like a Conclusion to me – I suggest you move it and end the Introduction with your research questions

Line 242: Umm, so how often were those involved also visiting the horses outside of the EAA sessions? Some more than others, some forming bonds with them?

Line 248: The subjects are aware that you are testing HRV? What were they told? Were they trying to synchronise their breathing etc? If so, horses breathe slower than us, therefore if their breathing slows it could also have a similar HRV effect – could this be done equally effectively with an app or guided meditation instead of EAA?

Line 252: comma missing after ….. over, consented…..

Lines 254-257: belongs in data analysis section

Lines 262-265: How did you try to separate this meditation-type exercise with touching the horses – a control could have been used here

Line 271: this requires TIMES for and between each event

Line 284: define ‘safety sentinel’

Line 285: more than one horse? Both?

Line 286: who are the ‘students’?

Line 288: define ‘heart breathing’

Line 289: define mutual choosing

Line 289: define hand grooming

Line 296-8: I’m not sure why this was done. Surely the horse continues to wear the monitor girth and thus continues to collect data without the need for invasive action or redundant measures.

Line 302: I suggest you put the participants information before the timeline – tell your reader ‘who’ before ‘how’.

Line 326: Familiar with – do you think this might have impacted the results?

Line 334: This needs to go earlier

Line 338: Put the number of horses used in the experiment earlier, in the Abstract

Line 345: How was ‘willingness’ assessed?

Line 348: How was ‘enjoyment’ assessed?

Line 350: How was ‘interest in interacting’ assessed?

Line 379: How long were the horses confined to stables before the interaction?

Line 385: I don’t understand why the Polar HR monitor would have been removed or the need to combine the post and pre measurements. How long was there between experimental phases? What else was going on in the surrounding area between these times?

Line 412: explain ‘heart breathing’

Line 412: horse or human ‘choosing’? How did the horse indicate its choice?

Line 423: define mutual choosing

Line 426: define breath heart beams

Line 427: That’s a shame as I think you missed some useful information by not using this horse as a control

Line 437: What did this look like? Explain the ‘grooming’ process – did each participant do exactly the same thing? Scratching, stroking, patting, talking….?

Lines 458-469: this belongs in the data analysis section

Lines 464: ‘easy to interpret’? As compared to …..

Lines 473: Tell your reader which software was used

Line 510: I suggest you label the diagram a) b) and c)

Line 512: a full description of the figure is required and both axes require labels

Line 517: reference your table here

Line 527: expand the table caption

Line 581: define ‘hand massaging’

 Figure 2: extend caption, label parts with a) b) etc and make sure each axis is labelled (is RR representing HRV here?)

Line 589: explain the difference between SDRR and RR

Lines 591-96: This belongs in the Discussion

Lines 595-602: this did not occur so can probably be omitted

Line 616: for clarity, simply refer to the current results in this section

Line 646: ‘less relaxed demeanour’ – as assessed by?

Line 646: surely this is ‘history’ rather than breeding

Line 648: ‘loss of interest’ – as assessed by?

Line 658: please provide a full table caption

Line 710: I’m not sure what this means – more required

Line 718: A full caption is required

Line 722: A full caption is required

Line 737: Horse or human?

Line 746: Were the number of words used recorded? It would have been good to know if they were getting more ‘chatty’

Line 748: Was this measured/assessed/significant?

Line 755: Is ‘arousing’ the best word here?

Line 832: Only three horses – this needs to go into your limitations section

Lines 840-844: Are these lines in bold for a reason?

Line 888: The Payne study is 5 years old, has there been more since?

Line 906: What other effect might gestures have had? What impact might other movements have had?

Round 2

Reviewer 2 Report

Thank you for replying to my comments and the changes you made to the paper.